# Beyond the Numbers: Quantitative and qualitative analysis of quality of life after deep brain stimulation for OCD

Emily Hemendinger[1]*, John A. Thompson[1,2], Jennifer Fishman[1], Katie Sinsko[1], Hannah Gebhardt[1], Rachel A. Davis[1,2]

1 Department of Psychiatry, University of Colorado Anschutz Medical Campus, Aurora, Colorado, United States of America, 2 Department of Neurosurgery, University of Colorado Anschutz Medical Campus, Aurora, Colorado, United States of America

¤ 1890 N Revere Ct, Anschutz Health Science Building, Department of Psychiatry, University of Colorado Anschutz Medical Campus, Aurora, Colorado, United States of America
* emily.hemendinger@cuanschutz.edu

## Abstract

### Purpose

Obsessive-Compulsive Disorder (OCD) severely impairs quality of life (QoL) in many sufferers. Deep Brain Stimulation (DBS) can be effective therapy for OCD that is refractory to standard treatment options, which include medications and cognitive behavioral therapy/exposure and response prevention therapy. The purpose of this study was to acquire qualitative data about the impact of DBS on QoL for people who have treatment refractory OCD.

### Methods

Ten subjects who received DBS surgery and are receiving ongoing DBS program-ming participated in this study. Pre- and post-surgery assessments (Q-LES-Q-SF and Y-BOCS) were analyzed for 8 of the 10 participants, and narratives from all 10 subjects were coded for themes. To assess individual correlations, analyses were completed within each participant, using solely their data points.

### Results

In six out of 8 subjects, improved QoL scores correlated with a decrease in Y-BOCS score, although this was only found to be significant in 2 of the 6 subjects. Qualitative data revealed that pre-DBS QoL was generally poor, with patients experiencing impairment in at least one domain. Post-DBS, most patients described improvements in overall QoL, particularly in the areas of mood and general functioning. However, many continued to experience difficulties in the social/relationship domain, including difficulty with social skills and transition to independent living.

**Data availability statement:** All relevant data are within the manuscript and its Supporting information files.

**Funding:** The author(s) received no specific funding for this work.

**Competing interests:** Emily Hemendinger, Katie Sinsko, Jennifer Fishman, Hannah Gebhardt, and Dr. John Thompson declare that they have no financial or non-financial interests to disclose. Dr. Rachel Davis provides paid consultation for Medtronic. This does not alter our adherence to PLOS ONE policies on sharing data and materials.

## Conclusions

These findings suggest that focused interventions addressing social skills and independent living may enhance post-DBS quality of life.

## Introduction

Obsessive-Compulsive Disorder (OCD) is a mental illness that involves the experience of both obsessions (intrusive, unwanted thoughts or images) and compulsions (physical and/or mental actions done in response to obsessions in attempt to reduce distress) [1]. On average, about 1 in 40 adults, or around 8.2 million adults, in the United States will develop OCD in their lifetime [2]. Additionally, OCD has been found to be the fourth most common mental illness, globally impacting 2–3% of the world's population [3].

This disabling condition leads many sufferers to experience a lower quality of life (QoL), with difficulty functioning in a variety of life domains including general, physical, psychological and social domains [4–8]. Specific factors that are associated with low QoL in those with OCD include psychiatric comorbidities, low perceived and actual social support, adverse effects of medication, effort expended engaged in severe obsessions and compulsions, social stigma, and strained relationships, especially in the context of accommodation (i.e., when others participate directly in or are involved in a person with OCD's compulsions, rituals, and/or avoidance) [4,5,8].

First-line treatment for OCD includes Exposure and Response Prevention (ERP) therapy and medications that target serotonin receptors, primarily Selective Serotonin Reuptake Inhibitors (SSRIs) [9–11]. Additional standard treatment often includes adjunctive therapies such as Acceptance and Commitment Therapy (ACT), Dialectical Behavior Therapy (DBT), Cognitive Therapy (CT), Mindfulness-Based Therapy, and augmenting medications [11]. While a combination of these treatments has been shown to be effective in symptom reduction, 40–60% of patients do not respond fully to traditional treatments [12], and about 20% of patients continue to experience severe and persistent symptoms [13].

Deep brain stimulation (DBS) is a minimally invasive neurosurgical treatment option for sufferers with refractory OCD [13]. DBS has regulatory approval in the United States (Food and Drug Administration – FDA) for Parkinson's disease, essential tremor and epilepsy [14–16]. DBS is approved via a Humanitarian Device Exemption (HDE) for OCD [17–19] and dystonia [20], and OCD is the only psychiatric indication with FDA-approval.

DBS requires brain surgery to implant electrodes in the deeper structures of the brain. This surgery is done using a method called stereotactic neurosurgery, in which the surgeon uses brain imaging and three-dimensional coordinates to guide insertion of the electrodes. The electrodes are connected to a pacemaker-like device in the chest wall that delivers electrical impulses. Stimulation modifies abnormal brain activity that occurs in OCD, with the goal of reducing obsessions and compulsions [21,22].

When DBS is used for OCD, about 60% of treatment-refractory patients experience a good response, defined as at least a 35% reduction in Yale-Brown Obsessive Compulsive scale (Y-BOCS) [13]. Across several meta and systemic analyses, the mean reduction in Y-BOCS scores across all patients who received DBS was 40% [23–30], which is clinically significant [13]. While many of these analyses had high levels of heterogeneity in design, population, and intervention timelines [31], there does appear to be an overall reduction of OCD symptoms in patients with treatment refractory OCD [30,32]. Additionally, DBS often leads to significant improvement in global functioning [21,23,24,27,30,32,33], a reduction in anxiety [28,32,33], and an improvement in cognitive performance [28].

Many of the various quantitative measures used to study QoL in DBS for OCD, include similar QoL domains and consistently demonstrate OCD symptom reduction, mood improvement, and improvements in areas of overall functioning post-DBS [8,25,26,32–36]. Conversely, the social/relationship domain often remains impaired in OCD [33,36]. Symptom reduction alone, as measured by quantitative rating scales, fails to capture the full impact of treatment; focusing on quality of life and functional outcomes is essential to understanding the broader patient experience [29]. A holistic approach, including both objective and subjective data, yields more comprehensive information about functioning beyond symptom reduction, including external influencing factors, actual opportunities in a patient's life for achievement and functioning, and how the patient feels about their current life situation and functioning vs. societal expectations (e.g., sometimes patients are quite satisfied with a living situation and functioning level that society may deem inadequate or not appropriate). Focusing on QoL also may address common measurement distortions that arise from strictly quantitative data, such as the affective fallacy, which is the tendency for people to use their current and temporary affective state to make judgments about how satisfied they are with their lives in general [29] or hedonic adaptation, a psychological process where people adapt to new circumstances and return to the same baseline level of happiness [37].

Qualitative data fills in these gaps and allows us to better understand patients' own perceptions of mental health and life changes. Qualitative data in the existing literature on DBS for OCD reflects similar improvements in OCD symptoms, mood, and anxiety [38–41] and maintenance of improvement when combined with ongoing therapy and support (e.g., exposure response prevention) [38,42]. Additionally, patients may experience improvement in perception, cognitive flexibility, and confidence [38,39,41]. Many patients also experience ongoing difficulties in certain areas. Specifically, factors that contribute to ongoing lower QoL include burden of normality (i.e., the difficulty individuals have transitioning from a state of illness or disability to a state of improved functioning or normalcy) [42], social difficulties, and adjusting to being symptom free [36,38,41,43].

## Purpose

This study aimed to acquire qualitative data on the impact of DBS on QoL in individuals with treatment-refractory OCD to inform and individualize treatment. We wanted to also explore the long-term changes and what factors were associated with long-term improvements (e.g., participating in therapy). This research adds to the current literature on QoL and DBS in OCD. Previous research has not focused on gathering data about the follow-up care being provided in addition to the DBS therapy. Additionally, previous studies have only focused on shorter time-frames post-DBS surgery, whereas this study collected quantitative data as far back as 2015 and qualitative was collected from patients who have had their DBS programming being done for over 10 years.

This study was guided by the following research questions: (1) Is QoL in those with OCD impacted by DBS? (2) Are there differences between quantitative data vs. qualitative data around symptom reduction and QoL? and (3) Are there QoL areas that remain problematic or unaddressed after surgery and ongoing programming? We examined the following hypotheses: (1) Patients with treatment refractory OCD who received DBS at least one year prior to the onset of this study will have experienced an improvement in QoL, as measured by the Quality of Life Enjoyment and Satisfaction Questionnaire Short Form (Q-LES-Q-SF) assessment, and OCD symptom reduction, as measured by the Y-BOCS. Across

patients, higher quality of life scores will be associated with lower Y-BOCS scores. (2) Patients will report similar improvements in QoL themes, specifically around improvement in OCD symptoms of QoL.

## Methods

### Participants

The OCD Surgical Program at the University of Colorado Anschutz has had a total of ten patients who have received DBS for treatment refractory OCD starting in 2015. There was no formal recruitment for this study as all patients were given the opportunity to participate in this study and all 10 consented to participate. While there was no formal recruitment, patients within the program needed to have met the following criteria before receiving surgery and the following programming:

- At least 18 years of age

- Severe to extreme OCD (Yale-Brown Obsessive Compulsive Scale score of 28 or higher)

- Has participated in at least 20 sessions of exposure and response prevention therapy with an OCD specialist

- Has tried at least three serotonergic medications (such as SSRIs) at FDA-max dose or higher for at least 12 weeks each at max dose. One of these agents needs to be clomipramine with a therapeutic blood level for 12 weeks.

- Has taken a long-acting benzodiazepine in addition to a serotonergic medication for at least a month

- Has taken an antipsychotic (dopamine blocking medication) in addition to a serotonergic medication for at least a month

- Does not have any active substance use disorder

- Does not have a severe personality disorder

- Does not have a primary psychotic or bipolar disorder

- No imminent suicidality (no recent or current suicidal behaviors or plans)

Quantitative data collection is a standard part of each participants' treatment; thus the data for this study was collected between the day of DBS assessment throughout present day. Participants consented to treatment before beginning with the assessment. Each participants' initial and follow-up assessment dates were different and ranges in day/month/year of start date vary, however the span ranges from August 2015 through June 2024. Participants were informed of the additional qualitative data collection in June 2023 and were given the opportunity to participate in this additional aspect from June 2023 until December 2023.

Each of these participants was diagnosed with treatment-refractory OCD and received deep brain stimulation (DBS) surgery as part of their clinical care via a Humanitarian Device Exemption. Eight out of ten of the participants are currently seen for ongoing clinical assessments by DBS Coordinator and Principal Investigator and for DBS programing by Co-Investigator. Two enrolled participants had their quantitative data excluded due to incomplete data collection, as they received their DBS surgery and pre-surgery treatment outside of the University. The remaining 8 participants were 20% female and 80% male. All participants in this study were evaluated at least one-year post-DBS surgery and initial programming.

### Consent and data collection

This research was reviewed by the Colorado Multiple Institution Review Board (COMIRB) and the board determined that this study was exempt in June 2023. This was deemed exempt because all participants were and are current patients of the Neuromodulation Program for OCD. Participants' involvement in this study did not impact their treatment. COMIRB determined that informed consent was not required for the collection of quantitative data as it was standard

of care. However, COMIRB did require that participants complete informed consent for the additional qualitative data that this research aimed to collect, as well as informed consent for the quantitative data to be collected for research and publication.

Informed consent was obtained from all individual participants included in the study. Investigators verbally reviewed informed consent paperwork with participants and participants signed paper copies of informed consent. Participants all provided informed consent and consent to publish all quantitative and qualitative data. The individuals in this manuscript have given written informed consent (as outlined in PLOS consent form) to publish these details.

Quantitative assessments and review of this data in participants' electronic medical records are standard of care for DBS treatment. Participants all provided informed consent which allowed the research team to access these assessments in their medical records and use them for research. Quantitative data from DBS assessments was accessed in medical records between June 2023 until August 2024. This qualitative data was not part of participants' medical records.

All data was only pulled and stored after informed consent was provided. Patients' charts are in a protected and encrypted EMR system (EPIC). PI (EH) and investigator (RD) were the only ones to pull the quantitative data from the patient charts after informed consent was signed. PI and Co-Investigator had preexisting treatment relationships with all patients and were the only ones who handled the data before it was coded. The data was coded and placed on a protected and encrypted Excel spreadsheet after it had been retrieved from the electronic medical record. The qualitative interview responses were linked to the unique identifiers at the time of collection and transcription. All information was kept in an encrypted database only accessible with a password by the research team.

The study was officially closed by the research team in August 2025.

## Materials

A modified version of the Quality-of-Life Enjoyment and Satisfaction Questionnaire Short Form (Q-LES-Q-SF) was used to collect qualitative data about quality of life (QoL) in each of the participants. The Q-LES-Q-SF [44] is traditionally applied as a quantitative measure and consists of a 5-point rating scale used to measure subjective ratings of an individual's QoL. In this study, the 15 QoL-related domains within the Q-LES-Q-SF were isolated, and participants were asked to provide descriptive, rather than numerical, responses in an interview style. The domains included were overall life satisfaction, physical health, mood, household activities, social relationships, work/school, family relationships, hobbies/leisure, daily functioning, sexual drive/interest/performance, economic status, living/housing situation, feeling dizzy/unsteady/falling, vision, and overall wellbeing. Alongside this qualitative data, the Q-LES-Q-SF and Yale-Brown Obsessive Compulsive Scale (Y-BOCS) were collected. The Y-BOCS [45] is a 10-item rating scale used to assess symptom severity and presentation in OCD. Lower scores on the Q-LES-Q-SF reflect lower QoL, and higher scores on the Y-BOCS reflect higher symptom frequency and impairment from OCD.

## Procedures

All patients treated in the OCD Surgical Program receive validated clinical rating scales prior to most programming appointments. The two measures examined in this study (Q-LES-Q-SF and Y-BOCS) are part of these regular assessments. The Q-LES-Q-SF and Y-BOCS were administered one to three times prior to surgery (if the participant was enrolled in the OCD Surgical Program before surgery), one time between lead implantation and pulse generator placement surgeries, and again before the first day of programming. These measures were then administered weekly for about one month, then every other week for a month, then monthly, with extending duration between assessments over time based on visit frequency as clinically indicated for each participant.

Change in Y-BOCS and Q-LES-Q-SF as a function of DBS therapy was measured by deriving the pre-therapy baseline from visits collected prior to DBS surgery. The number of pre-surgery visits from which the baseline was calculated varied between 1–3 sessions. Baseline Y-BOCS was defined as the mean of all available pre-DBS assessments to provide a

representative estimate of pre-surgical symptom severity. In treatment-refractory OCD, symptom severity can fluctuate over time due to intrinsic variability and ongoing treatment adjustments, and a single pre-operative timepoint may reflect a transient clinical state rather than typical illness severity. Averaging across multiple pre-operative measurements reduces the influence of short-term variability Using the pre-surgery baseline, subsequent follow-up sessions following the initiation of DBS therapy were compared to the pre-surgery baseline using percent change independently for both Y-BOCS and Q-LES-Q-SF. As subjects varied in the number of follow-up sessions over the course of 974–3203 days after stimulation was turned on, non-parametric summary statistics were derived based on quartiles of follow-up sessions across subjects. To account for the within-subject clustering and potential autocorrelation inherent in repeated measures, we analyzed the association between Q-LES-Q-SF and Y-BOCS using a linear mixed-effects model (LMM). The model included a random intercept for each subject and an autoregressive order of 1 (i.e., AR(1)) continuous correlation structure to account for the days post-stimulation. The program "R" (v4.4.3) was utilized to create the images presented in the results section.

The Q-LES-Q-SF interviews were conducted with ten out of ten participants who had received DBS surgery and programming with the OCD Surgical Program. These interviews occurred once per patient throughout the summer and fall of 2023.

For all participants, the Q-LES-Q-SF interviews and quantitative measures were administered by the DBS Coordinator and Principal Investigator. Qualitative data was organized and initially coded by DBS Coordinator and Principal Investigator. However, for a more unbiased opinion this data was later analyzed separately and coded for themes by three research assistants on the study team. Each research assistant independently reviewed the qualitative interviews completed by each participant. Each research assistant then created sets of inductive codes relating to content within the interviews, involving drawing connections between data points and identifying patterns [46]. Each coder worked to combine and narrow down the codes through separate reviews. The coders then met three times with each other and engaged in thematic analysis, a flexible framework used to compare codes and patterns, search for and review themes, define and name themes, and conclude with a final set of themes [47]. Coders reviewed the responses endorsed and codes created by reviewing the data from the transcripts together. Final codes were created, and themes were drawn from final review. The data was categorized into Pre-Surgery and Post-Surgery themes to compare reports of life experience before and after the procedure. Other theme categories identified included remaining symptoms, ongoing issues participants experienced post-treatment, and possible salient outside factors that more than one participant reported experiencing. The programs "Canva" and "Google Sheets" were used to create the figures presenting the qualitative data.

## Results

Summary evaluation of the percent change in Q-LES-Q-SF and Y-BOCS post onset of DBS therapy demonstrated that the majority of subjects evinced an improvement in quality of life and reduction in OCD severity (Fig 1). We sought to determine whether improvements in these two scales correlated at the individual subject level. Accounting for within-subject clustering and potential autocorrelation as a function of repeated measures, we applied an autoregressive corrected linear mixed model to association between change from baseline in Y-BOCS and Q-LES-Q-SF. The analysis confirmed a significant negative association between Y-BOCS severity and quality of life ($\beta = -0.15$, SE $= 0.04$, $p < .001$) with individual patient slopes that ranged from 0.106 to $-0.87$, indicating variability in the strength of the coupling between mood and OCD symptoms and substantial inter-individual heterogeneity in the underlying relationship. This suggests that while improvement in quality of life generally tracks with symptom reduction, the strength of this coupling is patient-specific (Figs 1 and 2). The rate of change in both the Q-LES-Q-SF and Y-BOCS were quite variable, and the patient-specific nature of this change can be further seen in Fig 3. In this OCD-DBS cohort, at final follow-up, the overall improvement in QoL was 43.48±39.85% (measured as percent change from baseline).

Analysis of the qualitative data from the Q-LES-Q-SF revealed several themes across participant outcomes. Many participants reported their pre-DBS experiences as "terrible," "bad," or "horrible," noting that their OCD symptoms occupied

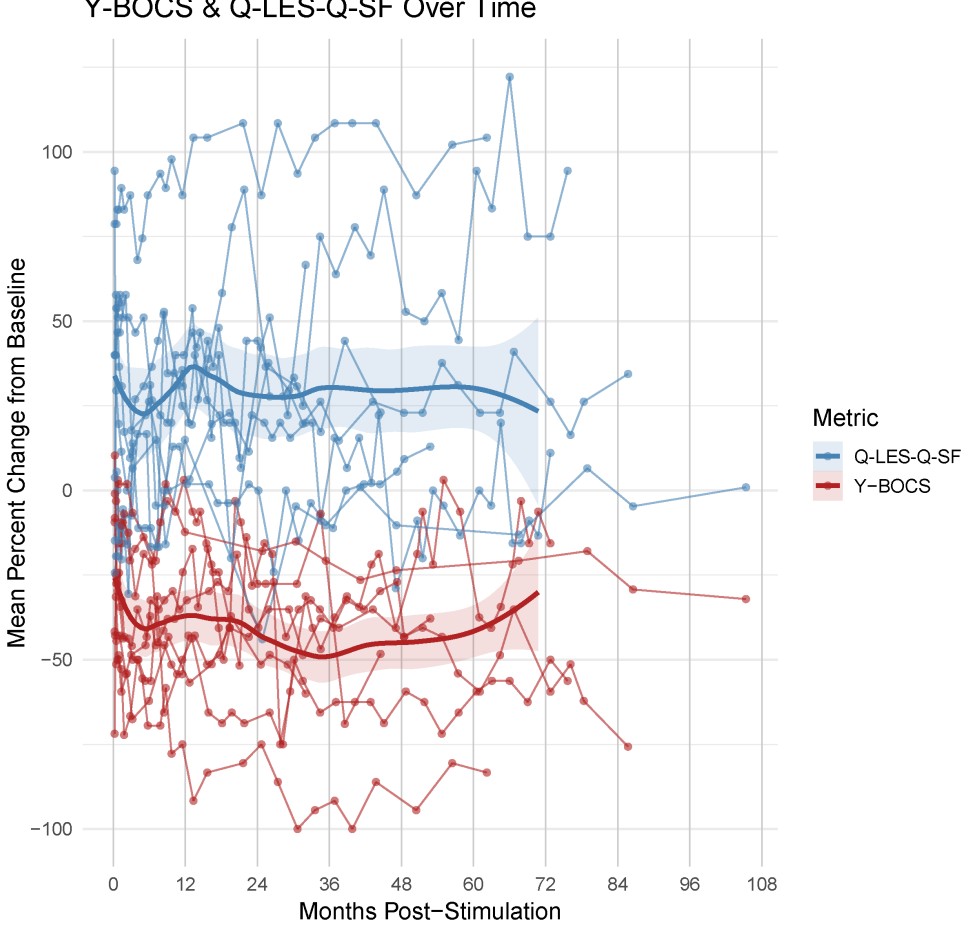

**Fig 1. Time course of change in Y-BOCS and Q-LES-Q-SF across subjects: The range of duration for the follow-up period varied across subjects (maximum range: 974-3203 days post initial stimulation).** Individualized trajectories of percent change in Y-BOCS and Q-LES-Q-SF were represented over the available follow-up period. Individual dotted lines denote unique subjects. The two thick lines are the linear least square regression (loess) for each metric across all subjects. LOESS data are modeled up to 72 months follow-up across subjects.

considerable amounts of time and inhibited their motivation to engage in activities of daily living, relationships, work/school, and hobbies (Figs 4 and 5). This was reflected in baseline QoL scores, with nine of ten participants endorsing QoL being affected in at least one area due to OCD symptoms.

At the time of the interview, eight of ten participants reported that life improved following the DBS procedure (Figs 4 and 6). The most notable changes were mood improvements, improved overall functionality, increased functionality around household tasks, slightly increased productivity at school or work, more engagement in hobbies (due to increased interest and increased energy), and decreased time spent on obsessions and compulsions (Fig 4 and Table 1). Eight of ten participants noted a reduction in frequency and intensity of compulsions. One participant described, "Yeah, my OCD is really good right now. You know, the obsessions aren't...I don't think about them that much. They don't come around as often." Another noted that they were "not as constricted by OCD [anymore]." While participants frequently noted improvements in family and social relationships, they also expressed a desire for more social connection and ability to feel more comfortable in social settings. One participant noted, "That's the part that still gets me a little bit.... I just wish I had a little bit better of a social support." Other participants echoed this sentiment. Although participants felt better equipped to establish

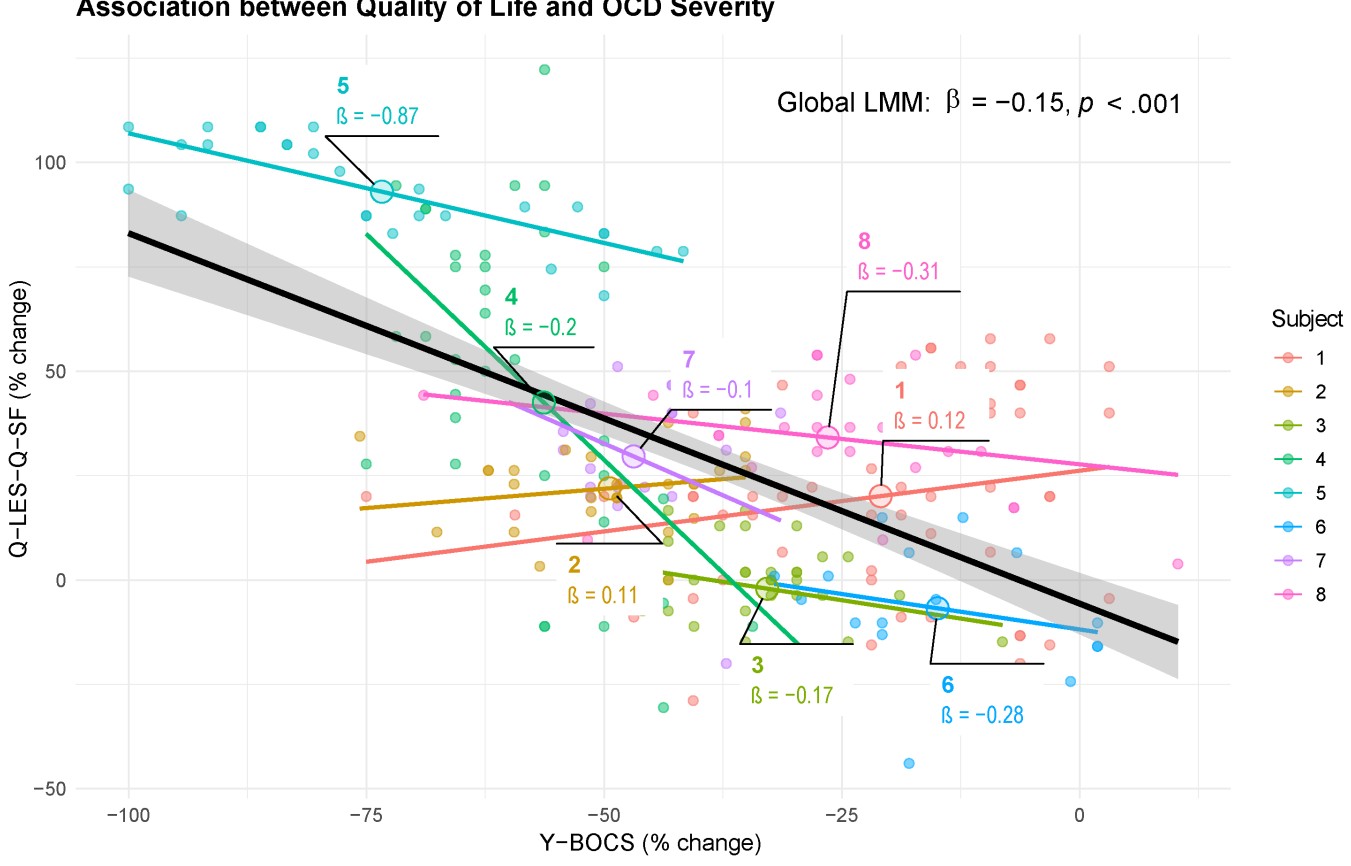

**Fig 2. Association of Y-BOCS and Q-LES-Q-SF change during DBS follow-up: Using a linear mixed model, we observed a significant association between quality of life and OCD severity, finding that for every 1-point increase in QOL (Q-LES-Q-SF), OCD symptoms (Y-BOCS) dropped by 0.15 points.** The negative relationship supports the conclusion that as Q-LES-Q-SF improves there is a concordant improvement in Y-BOCS.

and maintain boundaries with others and prioritize their own needs following DBS, they also perceived these changes as contributing to smaller social circles and a diminished sense of overall social support. There were two individuals that reported improvements in their living situations (due to seeing more functionality following DBS). These participants namely noted improved functionality with ADLs in their living environments (Table 2).

Other factors that improved QoL that participants shared are highlighted in the following quotes:

- "I think a big part was just having like so much contact with the team, honestly, and just like having that outside support because like, it can be so hard and isolating. It was also nice to get an outside perspective."

- "Volunteering helped. It led to me getting a job. That and going to church helped with socializing."

- "Being able to set boundaries and not engage with friends who don't respect me."

- "Advocacy work has been really satisfying; helping others and connecting them with resources."

In addition to specific areas of change post-DBS, participants were asked about the timeframe of changes, as well as if changes felt long-lasting. 9 out 10 participants noted immediate improvement in their mood, stating that the improvement occurred the same day as the first programming session or within the first several weeks of programming. Eight out of 10

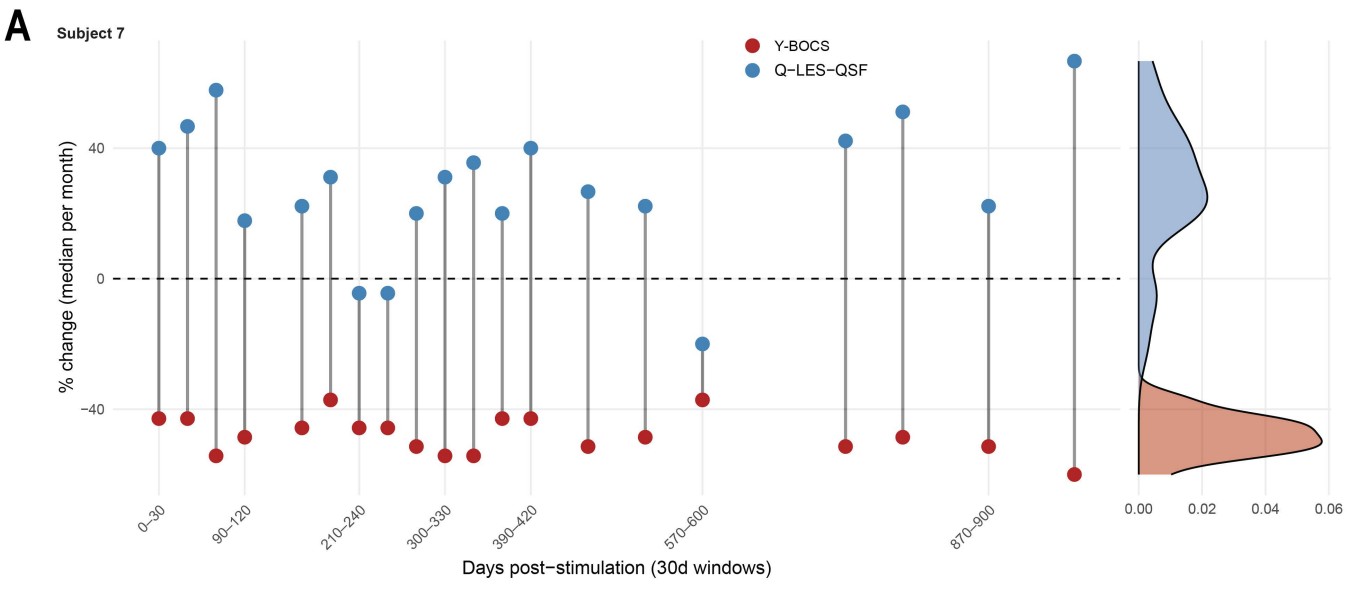

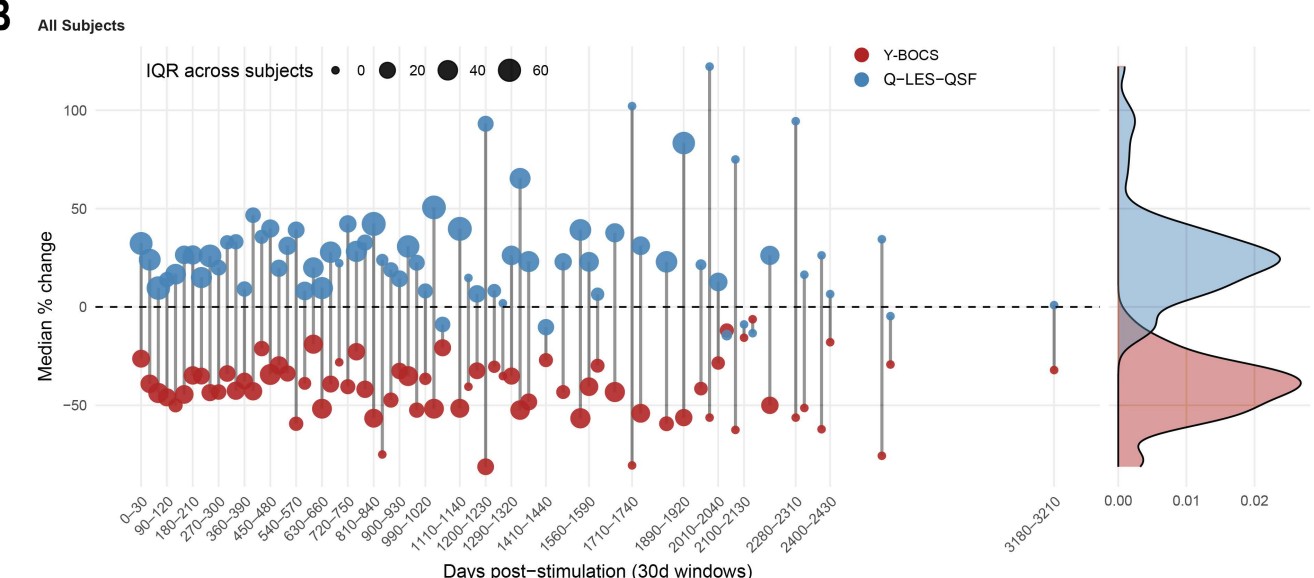

**Fig 3. A. Representative Subject (7) showing the change overtime for Y-BOCS and QoL at 1 month resolution.** It shows that the QoL is more variable than the YBOCS. **B.** All subjects – same resolution (each circle size represents the IQR (interquartile range) for all subjects at that time point. Each dot size conveys the number of subjects which will influence IQR). This is a higher resolution plot in 30-day medians.

participants noted that their OCD symptoms improved with the first 3–6 months. Finally, 5 out of 10 participants reported long-lasting improvements from DBS, while 3 out of 10 participants noted partial long-lasting improvements, with partial symptom return and 2 out of 10 participants reported feeling no effect from DBS anymore. A summary of each subject's responses for timeframe and longevity of DBS impact can be found in Table 3.

During the interviews, a trend emerged where the participants with higher reported symptoms and lower quality of life, were hesitant to share their experience about DBS. One participant stated, "I'm not sure you want me for this research.



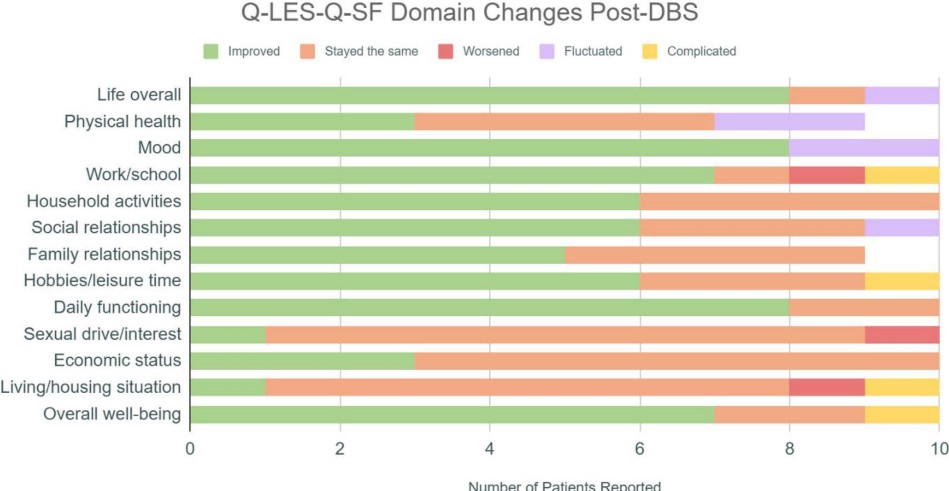

**Fig 4. Self-reported changes regarding quality-of-life domains post-DBS: In interviews, patients were asked about each domain in the Q-LES-Q-SF scale to understand whether quality of life stayed the same, improved, or worsened because of DBS surgery.** The figure depicts patients' responses to the interview questions, including two new categories of "fluctuated" and "complicated," to include patients who felt that their experience did not fall under "improved," "stayed the same," or "worsened.".

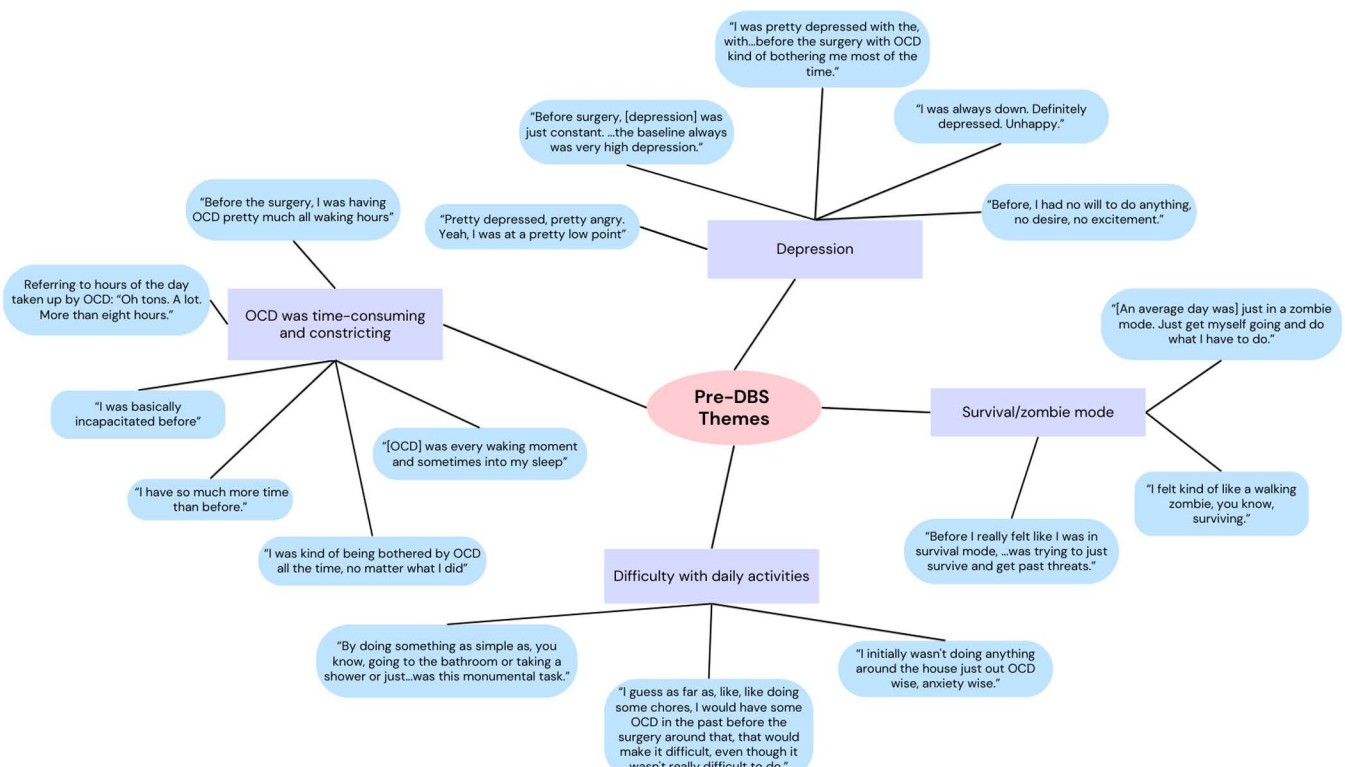

**Fig 5. Visualization of qualitative coding process for pre-DBS themes: Select patient quotes from interview transcripts were used to illustrate the qualitative coding process.** The figure does not include all pre-DBS themes or every patient quote associated with each theme.

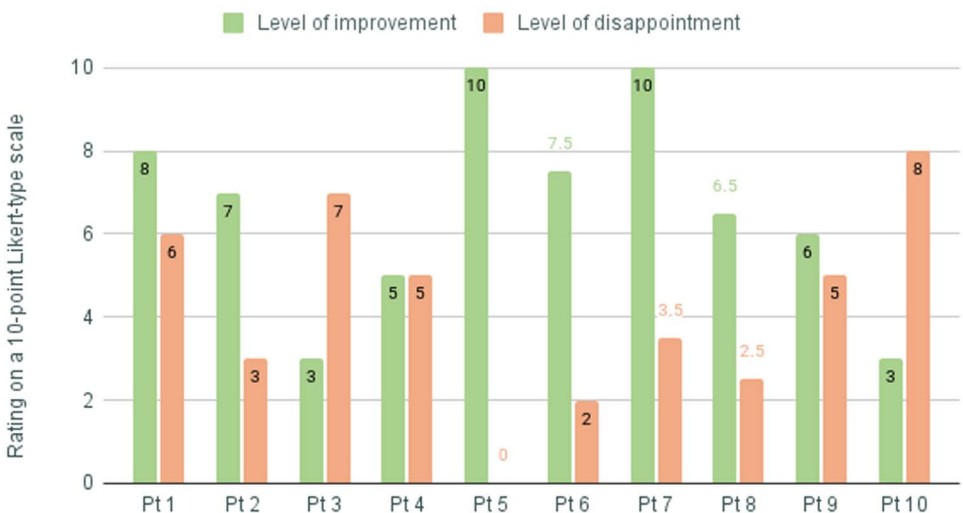

**Fig 6. Self-reported ratings of improvement and disappointment in quality-of-life post-DBS:** The figure displays patient responses to the interview questions: "Where would you rank the level of improvement in the quality of your life since receiving DBS surgery on a scale of 0-10, where 0 is not improvement in quality of life and 10 is the tremendous improvement in quality of life?" and "Out of all the domains on the Q-LES-Q-SF, were there any that stayed the same? If yes, are you disappointed that things haven't changed? Where would you rank that level of disappointment on a scale of 0-10, where 0 is not disappointed at all and 10 is the most disappointed you've ever felt?".

**Table 1. Participant quotes indicating improvements in functioning.**

| |
|---|
| "Before [DBS], I had no will to do anything, no desire, no excitement. And now I feel a bit more like the so-called normal human that can actually, like, feel things and live life being excited about things." |
| "Before DBS I was just surviving, like, I wasn't doing any hobbies because surviving was all I could focus on…I was just there, kind of sitting. So, after…I've been able to do a lot more that I've always wanted to do but never had the ability to mentality handle." |
| "I have so much more RAM or bandwidth in my head because it's not full of obsessions. And so, I'm trying to fill that with productive activities and social support and meaningful things; I'm trying to build a life that's not consumed by OCD. That part is taking some work still, but my headspace is tremendously better. |

**Table 2. Participant quotes regarding housing and living environments.**

| Satisfied | "Since the surgery I can like...you know, I got my license after, my driver's license after I had surgery done. I can, you know, drive myself to the, to the grocery store and do something like that."<br>"I'm able to do the dishes now and help out around the house and feed the animals and play with my animals and that sort of stuff." |
|---|---|
| Unsatisfied | "So I'm sort of around my parents a little bit more. So that's makes it a little bit more difficult."<br>"I want to be basically doing the...involved in the basic fundamentals of life, like having a job and living on my own, or living with...having friends or a group of friends or something…"<br>"I love where I live now. I just hate living by myself." |

**Table 3. Summarized participant responses on timeframe and longevity of DBS impact.**

| Subjects | Timeframe of Response | Longevity of DBS Impact | Quotes |
|---|---|---|---|
| Subject 1 | Immediate | Some long-term impacts, but feels effectiveness as decreased | "I feel better right away and one to three months in, I felt awesome mood wise and OCD wise." "Six months in, OCD and mood were getting worse and one year in things were not good." "Recently, my mood has been better. I'm more present. And, just able to not have as many obsessions, not think about it as much." |
| Subject 2 | Several months | Long-lasting | "It seemed better, you know, it seemed better, but I really...I really couldn't, I couldn't quite tell after a month of having that put in me yet, you know, after a month yet. And so, I think it, I think it took a while, several months before I really saw the difference, if you know what I mean. After six months, I was a lot happier."  "Yeah, there's been a lot of times where I wasn't doing my therapy and, and that would bring back the symptoms, but, you know, not that many, I'm still real happy, you know. |
| Subject 3 | Mood: immediate, OCD: around 6 months. | Long-lasting | "Oh I definitely have an answer for this. Depression, almost immediately. OCD about six months" "Things have definitely decreased overall, but I have my ups and downs. Moral of the story. I think without DBS, I probably would be dead right now." |
| Subject 4 | Immediate | Long-lasting | "Immediate, right after it was turned on. Happiness right away." "Things feel stable. I finished school and just got a job." |
| Subject 5 | Immediate | Long-lasting | "Yeah, I think it was pretty much, pretty much right away." "Over time it's been steady improvements and now I feel stable." |
| Subject 6 | Around 3–6 months | Reports not feeling much effect anymore. | "It took time. Maybe 6 months. My mood was definitely better after like 3 months." [In the first year] "I spent a lot of time in treatment for eating disorder. So like the eating part was still pretty bad. But I didn't struggle as much with like the low, low mood and the anger outburst I kind of had had beforehand. So, that was good." "I got much better 4-5 years after. Then recently the eating disorder got worse, my mood got worse and the OCD got worse. I don't feel much effect anymore." |
| Subject 7 | Immediate | Long-lasting | "Yeah. Mine luckily was very fast. I started feeling…on the first session I felt a little bit better. That second day we picked settings and I was like a lot better. And then that third day, I just felt so much better." "The OCD is definitely present still. But everything is decreased. And now this next phase is, how do I build a life that's compatible with what I want?" |
| Subject 8 | Within the first 2–3 weeks, primarily mood. OCD first several months. | Some lasting but feels effectiveness has decreased. | "Well, after a couple weeks my mood...2 or 3 weeks, my mood was a lot better." "I'm able to take classes, but yeah I feel like it's stopped working as much." |
| Subject 9* | First several weeks | Somewhat lasting, but tics and mood have not had lasting impact. | "Yeah I noticed improvements within the first week to several weeks." "My life isn't where I want it to be. My depression and tics are still bad. I think DBS helped a little." |
| Subject 10* | First month | Some improvement initially, but over time feels it has worn off. | "After one month, I felt pretty good. But my eating disorder was still bad." "After six months, the changes felt fewer and more subtle. But after one year I felt better and had more freedom." "Like, I felt good and then it like, totally wore off. Not that the settings don't do anything, but like, I don't feel the changes anymore." |

*Subjects 9 and 10 were the subjects who were excluded from quantitative analysis due to missing data from pre-surgery.

I'm not a very good example of DBS helping right now." During the interview, the same participant also noted, "I know my current state is not a reflection of DBS. There are a lot of external factors that are impacting me and my life." The rigidity and lack of insight into the bigger picture led 6 out of 10 of the participants to initially state that either their quality of life was poor or had not changed. However, by the end of each interview, participants each noted the improvements in their overall symptoms and lives. This can be seen in the following quotes from the same participant:



- "Ok, yes. DBS improved some areas [of my life]. My family relationships were definitely improved. I involved them more in my life which improved our relationships."
- "I mean, after [DBS], I feel like I probably...did have a better outlook on life... Well not probably. I did have a better outlook on life."

Other patients with similar initial reactions had comparable sentiments (Table 4).

## Discussion

Our qualitative data provides a more robust picture of quality-of-life improvements than captured in quantitative measures alone. Specifically, patients note that DBS has given them a new baseline and expanded their belief in self-capability, as well as allowing them to consider life opportunities they hadn't previously thought possible (e.g., school, relationships, or hobbies). Therefore, quantitative measures may seem to not improve as much because the range of life opportunities has expanded for our patients post-DBS, i.e., the scale has been reset.

Since DBS therapy for OCD does not address external stressors, these lived experiences highlight the importance of comprehensive post-treatment care. A narrow focus on DBS therapy alone will overlook key factors that influence long-term outcomes and QoL. These findings inform future recommendations for clinical care and support following DBS, and potentially even prior to DBS surgery, during the preparatory phase. While DBS can be an effective intervention to treat severe and refractory OCD, it should not be a stand-alone treatment.

In our research. following up with therapy (typically Exposure Response Prevention therapy) and appropriate medication management were crucial tools in managing symptoms during long-term follow up. Several participants reported that, while DBS helped to initially mitigate OCD symptoms, these effects diminished somewhat over time. In the quartile follow-up assessments, 4 participants noted the need for ongoing, additional individual therapy and medication management following DBS. One participant remarked, "[Since DBS, I've had] several changes to medications, primarily to combat refractory depression." Another participant noted, "I believe the DBS was the tool, but ERP is how I changed old habits." These reflections highlight the need for step-down care following DBS and the importance of reinforcing positive results from DBS through continued medication management and skills practice. However, while the importance of post-DBS ERP may be intuitive and shown in qualitative data, previous more controlled data did not unambiguously show this additional effect [48].

The participants who displayed lower levels of satisfaction post-DBS and had more difficulty sustaining improvement were often inconsistent with following recommendations for medication and therapy. They were more likely to hope that DBS alone would manage their symptoms. Participants who had more realistic and balanced expectations of the role of DBS post-surgery were more likely to be active and engaged in their life and treatment. These participants were better

**Table 4. Participant quotes regarding big picture improvement.**

| |
| --- |
| "My physical health is a mess…I regularly have 2-3 days a week where I have suicidal ideation and depression… Honestly, I think the biggest thing that has changed for me is that before DBS, I had no hope and like, every day was just kind of a waiting to die situation where now I actually have hope for my future and what I'm going to do with my life and that's nice." |
| "Yeah. [My Q-LES-Q-SF score] hasn't increased since surgery. And I'm like, I think it's just because I'm doing life in a different lens that, like, my life is much better, but I'm viewing it at a harsher lens now where I'm like, oh yeah, I'll give it the score and this score, where before I was like, grinning and baring it I suppose? So. I'd say my life satisfaction has improved in terms of...I just feel happier and more alive." |
| "And so trying to fill that with productive activities and social support and meaningful things is trying to build a life that's not consumed by OCD. I think that part is lagging and still taking some work, but my headspace is tremendously better." |
| "It's easy to get caught up in weeds sometimes in the day to day and not remember...And so you really focus in on what isn't going well. And so when you take a step back and like compare pre to post I think our brains do a really good job of forgetting how bad things used to be as a survival mechanism and when you really just think about it and all the like good things going on like, I'm in a good spot." |

able to recognize the limitations of DBS and how DBS was not a be-all-end-all. These participants were more likely to be active in either therapy, medication management, holding an occupation or attending school, engaging in vocational training, engaging in social skills training, or actively improving their relationships.

External factors, such as psychiatric and physical co-morbidities, housing, social support, grief and loss, and family dynamics also impact mood, OCD symptoms, and functioning. While DBS cannot control external factors, it is necessary to recognize that external factors can still have a large impact on perceived QoL. It is important to assess a variety of areas pre and post DBS treatment, including symptoms, functioning domains, and insight [49]. When considering how responsive a patient is to DBS. Without considering these factors, a patient may be incorrectly deemed non-responsive to the treatment.

Several participants experienced fluctuations in mood during DBS therapy and attributed these changes to external factors, including symptom status of co-morbidities, physical health issues, financial status, and relationships, rather than related to DBS. The two participants that reported the least improvement in the qualitative interviews currently have active eating disorders. These results are consistent with previous research demonstrating the significant negative and long-lasting impact eating disorders have on quality of life and functioning [50–52]. Despite lower improvement scores and higher levels of disappointment with DBS, both patients did acknowledge that they would not want to change their decision to receive DBS and that DBS has improved at least one area of their lives. If our team was not doing thorough assessments and gathering qualitative data, these patients may have been deemed non-responsive to DBS.

Patients may retrospectively recognize that their mental state pre-surgery may have interfered with their ability to fully understand pre-surgery psychoeducation about expectations post-DBS [41,53–57]. Additionally, programming with patients who have low insight may lead to motivation, ability to recognize symptom changes, and may change long-term goals and expectations across individuals. In our research, this was evident during the qualitative interviews as discussed in the results section and Table 3.

In order to accurately acquire the most comprehensive picture of DBS responsiveness and current QoL, thorough assessment should be completed around level of insight prior to surgery and post-surgery. Expectations for DBS treatment, co-morbidities, and external factors (e.g., support system, financial status, occupational status, etc.) pre- and post-surgery should also be evaluated.

Additionally, the psychiatrist adjusting the DBS stimulation parameters should use objective clinical observations about patient improvements, as well as permit family and friends to participate in programming sessions to help with assessment and to allow patients to feel more comfortable [57]. Whether the patient has insight into their symptoms or not, having additional observations and data can give programmers and treating clinicians information to better address patient care. For instance, gathering information from family and friends may provide insight into accommodation levels and compliance with ERP homework and medication routines. It is also important to note that a patient's assessment of their own progress may be impacted by how they feel transitioning to a life without symptoms and what their role in their life and relationships may be. This "burden of normalcy", transition to a life post-OCD, and new identity (e.g., not being sick), as well as higher desire for socializing, but having a lack of social network or being out of practice, may also lead to the difficulties our subjects discussed in their qualitative interviews [36,38,43,56]. Potential treatment recommendations to address some of the ongoing issues may include occupational therapy, social skills training, and group therapy to address deficits experienced in executive functioning, relationships, and school/job functioning. Peer support groups for patients who have had DBS for OCD may also be a means for improving quality of life in the social domain [41,56].

Limitations of this study include the small sample size, absence of pre-DBS quantitative data for two who had DBS surgery at an outside hospital, and possible affective biases that may be displayed by patients. The interviews were conducted following surgery and programming, and participants were at different time points in their programming, ranging from 2 years to 8 years post-surgery. This may have led to skewed data due to memory biases and current life circumstances impacting how participants viewed the success of their treatment. Future research could include interviews and

assessments completed both pre- and post-surgery. Family members and caregivers could also be interviewed to obtain more robust information about patients' QoL and differences in experienced versus perceived QoL.

## Conclusion

It should not be assumed, based on quantitative measures alone, that QoL has not improved or has only minimally improved in participants with OCD post-DBS. Due to increased awareness of life opportunities, participants' subjective reports of improvement provide more accurate and useful perspectives. This is important because accurate perspective allows for more individualized and holistic interventions based on participants' life satisfaction and personal goals. DBS alone is not sufficient for improving QoL, and post-DBS treatment should include support around interpersonal relationship and social skills, executive functioning, employment, and the transition to more independent living. Managing expectations prior to DBS surgery often leads to improved insight post-DBS about the fact that DBS alone will not be sufficient for improved QoL, and participants with more insight are more likely to be active participants in their recovery.

## Supporting information

**S1 File. Summary data.**
(XLSX)

## Acknowledgments

The authors would like to acknowledge all the patients from the Neuromodulation and OCD Program who participated in this study. The authors are grateful for the stories and data that these patients provided. The authors would like to also thank Dr. Joseph Sakai for his help with conceptualizing the quantitative data analysis. The authors would also like to thank Hunter Orr for his work editing transcripts for translational related errors.

## Author contributions

**Conceptualization:** Emily Hemendinger, Rachel A. Davis.

**Data curation:** Emily Hemendinger, John A. Thompson, Jennifer Fishman, Hannah Gebhardt, Rachel A. Davis.

**Formal analysis:** Emily Hemendinger, John A. Thompson, Jennifer Fishman, Katie Sinsko, Hannah Gebhardt.

**Investigation:** Emily Hemendinger.

**Methodology:** Emily Hemendinger, John A. Thompson.

**Project administration:** Emily Hemendinger.

**Resources:** Emily Hemendinger, Jennifer Fishman.

**Visualization:** Emily Hemendinger, Jennifer Fishman, Katie Sinsko, Rachel A. Davis.

**Writing – original draft:** Emily Hemendinger, John A. Thompson, Jennifer Fishman, Katie Sinsko, Hannah Gebhardt.

**Writing – review & editing:** Emily Hemendinger, John A. Thompson, Jennifer Fishman, Katie Sinsko, Rachel A. Davis.

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
