## [Decision Letter · Decision Letter 0]

6 Jan 2026

PONE-D-25-56441Beyond the Numbers: Quantitative and Qualitative Analysis of Quality of Life After Deep Brain Stimulation for OCDPLOS One

Dear Dr. Hemendinger,

Thank you for submitting your manuscript to PLOS ONE. After careful consideration, we feel that it has merit but does not fully meet PLOS ONE’s publication criteria as it currently stands. Therefore, we invite you to submit a revised version of the manuscript that addresses the points raised during the review process.

In your revision, please provide a careful point by point response to all reviewer comments, as well as highlighting all required amendments within the manuscript. In particular, please add the data referred to by reviewer 1 (point 11) assuming that this is available, and confirm the ethical review process as highlighted in  points raised by reviewer 2.

We look forward to receiving your revised manuscript.

Kind regards,

Clare Eddy

Academic Editor

PLOS One

Journal Requirements:

“I have read the journal's policy and the authors of this manuscript have the following competing interests:

Rachel Davis provides paid consultation for Medtronic.”

3. We note that your Data Availability Statement is currently as follows: “All relevant data are within the manuscript and its Supporting Information files.”

Reviewers' comments:

Reviewer's Responses to Questions

**Comments to the Author**

1. Is the manuscript technically sound, and do the data support the conclusions?

Reviewer #1: Partly

Reviewer #2: Partly

2. Has the statistical analysis been performed appropriately and rigorously? 

Reviewer #1: No

Reviewer #2: Yes

3. Have the authors made all data underlying the findings in their manuscript fully available?

Reviewer #1: Yes

Reviewer #2: Yes

4. Is the manuscript presented in an intelligible fashion and written in standard English?

Reviewer #1: Yes

Reviewer #2: Yes

5. Review Comments to the Author

Reviewer #1: The authors describe quantitative (N=8) and qualitative (N=10) symptom and quality of life outcomes after DBS for TR-OCD. Combining qualitative and quantitative outcomes provide a fine-grained picture of the follow-up of this small cohort. While the manuscript is overall well-written and a potentially relevant addition to the literature, there are several issues that could be addressed to further improve the impact.

1. There are two recent (2025) meta-analyses on DBS for OCD that may provide a more complete and actual overview of the literature, which could be added to the introduction.

2. The authors do not cite the largest cohort of DBS for OCD published to date, which may be relevant to their introduction (Denys et al., Am J Psych, 2020, with a follow-up also on QoL in Graat et al., Biol Psych, 2021).

3. Line 185-187: The authors describe the calculation of a baseline Y-BOCS, but do not precisely describe how this calculation was performed. Did they take the mean of the 1 to 3 pre-DBS Y-BOCS values or did they perform a different calculation?

4. Line 189-190: the authors describe the number of follow-up sessions in days, which seems incorrect.

5. The authors describe and show that they divided the number of follow-up sessions in quartiles per patient (189-190 and figure 2). This may be somewhat misleading as the number of follow-up sessions differs between patients (973-3203). For example Q4 may include follow-up up to day 973 in one patient and up to 3203 in another. These large differences in follow-up in one X-axis label may distort interpretations. The authors may try to use a different X-axis labelling, e.g. months since start of stimulation.

6. Line 191-192, Figure 3: the authors use Pearson correlations to test the within-subject correlation between percent change in Y-BOCS and Q-LES-Q-SF. Given that these are repeated measures within subject over time, there likely is autocorrelation, which may influence the correlation. In addition, the authors now only tested the within-subject correlation. The authors may consider using linear mixed models to test the association taking autocorrelation into account and test significance across subjects.

7. Line 196-208: do the authors have references supporting the methodology to analyze the qualitative data?

8. The authors report on observational quantitative and qualitative data, did the authors consider the use of reporting guidelines such as STROBE and/or SRQR?

9. The authors report quotes in the discussion, which may be better seen as results.

10. The authors report several recommendations for clinical care based on their results. Given the small N and the observational nature of the data, the authors should tone down their tone and present their findings in the context of broader data. For example, while the importance of post-DBS ERP is intuitive and reported in the qualitative data, more controlled data did not unambiguously show additional effects (e.g. Graat et al., Psych Med, 2022).

11. Line 366-367: in their conclusion, the authors note “It should not be assumed, based on quantitative measures alone, that QoL has not improved or has only minimally improved in participants with OCD post-DBS.” However, the authors do not present data on the general (e.g. mean) overall improvement of QoL in their sample. The authors may consider adding this data, also including e.g. percentage of (partial) responders.

Reviewer #2: This is an important manuscript that discusses changes in quality of life after deep brain stimulation for OCD, as well as the importance of follow-up care for patients. Additional clarity about analyses, outcomes, and results would be beneficial.

Abstract:

Authors should clarify that analyses were completed within each participant, using solely their data points, in order to assess for individual correlations

Introduction:

The introduction, while it covers relevant information, is a bit choppy, with some very short (i.e. 2 sentence) paragraphs

The final hypothesis should be clarified- “Patients will report similar improvements in QoL themes.” Were there particular themes that the author anticipated would improve versus others?

Methods:

It should not be necessary to include specific numerical dates for initial assessments, date of ethics determination, closure date, and date of acquisition of quantitative data, particularly given the small sample size. Either year or month and year (without date) should be sufficient.

Authors should not include the coding system for their deidentification, as this is providing information to a global audience that theoretically could lead to patient identification.

On page 6, “quantitative assessments and reviews of this data” should be “quantitative assessments and review of these data.”

The method and related IRB approvals is unclear. If this is a retrospective analysis of already-collected clinical data, only a PHI waiver from the IRB should be necessary. However, it seems as if the qualitative analysis was added on and NOT part of the clinical procedures, just research. Thus, patients should be consented for this. The writing is not clear- it first states that no ethical approval was needed but then states that participants provided informed consent. Please clarify.

Given that individual correlations were conducted, how many data points (range across patients) were in the correlations?

Results:

Given that there is an extended period of follow-up per patient after DBS, authors should describe time line of improvements in YBOCS score and QoL For example, on page 9, authors note “Eight of ten participants reported that life improved following the DBS procedure.” What is the time frame on this? Did it persist? Were improvements in QoL and DBS based on last follow-up, any time point, etc?

Authors randomly put in that there were no changes in vision or dizziness following DBS- was that part of the QoL measure, or a statement about side effects in general?

Were the timelines of changes in YBOCS and QoL different after surgery? We may anticipate that recognition of QoL may be somewhat delayed as compared to symptom improvement, which may impact correlations. Was this examined?

Discussion:

Authors note “Patients may retrospectively recognize that their impaired mental state…post-DBS.” Are they indicating that there are ethical concerns about these patients receiving DBS? This is what this sounds like in reading this sentence.

Quotes should likely be in the Results section if they are part of a qualitative analysis; there are an extensive number of quotes in the discussion that may be better placed elsewhere.

It appears that there was a push from the investigators to have the patients assess their QoL in a different way; authors should be very careful about this, as this may be interpreted as the clinical providers placing undue pressure on the patients in order to say that their QoL was actually better than initially reported (e.g. page 13 “Only after building insight and being asked to zoom out to see the bigger picture…overall symptoms and lives.”)

Figures:

Figure 1 is unclear. It looks like all except subject 1 started with a change in YBOCS score to begin with? It also appears that most participants had an initial drop in YBOCS and then an increase later- how many patients were long-term responders? Similar questions for QoL. There is a similar question for Figure 2- for the first data point, was this after xx days of stimulation? Sounds like after a quartile of stim- which I assume could be very different depending on follow-up period?

Figure 3 is unclear. Is the y-axis change in QoL, and the x-axis YBOCS change? They don’t look like raw scores, but it is unclear. Also, if possible, scales across patients should be consistent.

6. PLOS authors have the option to publish the peer review history of their article (what does this mean?). If published, this will include your full peer review and any attached files.

Reviewer #1: **Yes:** Roel Mocking

Reviewer #2: No

---

## [Author Response · Author response to Decision Letter 1]

17 Feb 2026

Response to editor and reviewers:

Thank you for your feedback and careful review of our manuscript. We have completed all revisions and our responses to each can be found below. (These responses can be found in our separate document as well).

Editor

• A letter that responds to each point raised by the academic editor and reviewer(s). You should upload this letter as a separate file labeled 'Response to Reviewers'. Completed

• A marked-up copy of your manuscript that highlights changes made to the original version. You should upload this as a separate file labeled 'Revised Manuscript with Track Changes'. Completed

• An unmarked version of your revised paper without tracked changes. You should upload this as a separate file labeled 'Manuscript'. Completed

Journal Requirements:

1. Please ensure that your manuscript meets PLOS ONE's style requirements, including those for file naming. Completed

2. Thank you for stating the following in the Competing Interests section: “I have read the journal's policy and the authors of this manuscript have the following competing interests: Rachel Davis provides paid consultation for Medtronic.” This does not alter our adherence to PLOS ONE policies on sharing data and materials. Completed

3. We note that your Data Availability Statement is currently as follows: “All relevant data are within the manuscript and its Supporting Information files.” Additional de-identified data has been added to the supporting information files. See supporting information file for a summary sheet with days post stimulation, Y-BOCS and QoL percent change values. Completed

4. If the reviewer comments include a recommendation to cite specific previously published works, please review and evaluate these publications to determine whether they are relevant and should be cited. There is no requirement to cite these works unless the editor has indicated otherwise. Completed

Reviewer #1:

1. There are two recent (2025) meta-analyses on DBS for OCD that may provide a more complete and actual overview of the literature, which could be added to the introduction.

We agree that these references add a more robust and more current overview of the literature. These references and related information were added to the introduction and are listed under reference numbers [31] and [32]. Lines 83-89 (tracked changes manuscript).

2. The authors do not cite the largest cohort of DBS for OCD published to date, which may be relevant to their introduction (Denys et al., Am J Psych, 2020, with a follow-up also on QoL in Graat et al., Biol Psych, 2021).

We agree with the reviewer about the inclusion of these references, especially Graat et al., 2021. These references and related information were added to the introduction and are listed under reference numbers [30] and [33]. Lines 83-89 (tracked changes manuscript).

3. Line 185-187: The authors describe the calculation of a baseline Y-BOCS, but do not precisely describe how this calculation was performed. Did they take the mean of the 1 to 3 pre-DBS Y-BOCS values or did they perform a different calculation?

We calculated the average of the pre-DBS Y-BOCS baseline days. If only 1 day was collected during the pre-DBS period, then that day was used as the baseline.

4. Line 189-190: the authors describe the number of follow-up sessions in days, which seems incorrect.

This was edited for clarity and now appears in lines 216-217 (tracked changes manuscript)/210-211 clean copy: “As subjects varied in the number of follow-up sessions over the course of 974-3203 days after stimulation was turned on, non-parametric summary…”

5. The authors describe and show that they divided the number of follow-up sessions in quartiles per patient (189-190 and figure 2). This may be somewhat misleading as the number of follow-up sessions differs between patients (973-3203). For example Q4 may include follow-up up to day 973 in one patient and up to 3203 in another. These large differences in follow-up in one X-axis label may distort interpretations. The authors may try to use a different X-axis labelling, e.g. months since start of stimulation.

We have updated Figure 2 to include the size of each subject’s quarter percent change marker to account for number days during quarter. See new Figure 2.

6. Line 191-192, Figure 3: the authors use Pearson correlations to test the within-subject correlation between percent change in Y-BOCS and Q-LES-Q-SF. Given that these are repeated measures within subject over time, there likely is autocorrelation, which may influence the correlation. In addition, the authors now only tested the within-subject correlation. The authors may consider using linear mixed models to test the association taking autocorrelation into account and test significance across subjects.

We agree with the reviewer’s concern and have replaced the current analysis using Pearson correlations with a linear mixed model using Q-LES-Q-SF as a fixed effect, subject as a random intercept and included a continuous autoregressive correlation to account for time between measurements. The LMM confirmed a significant association between symptom reduction and quality of life improvement (β= -0.15, p < .001), demonstrating that the relationship remains robust after accounting for the repeated measures structure and autocorrelation. We have updated the Methods and Results sections (Lines 219-223 tracked changes manuscript/212-216 clean copy) to reflect this more rigorous analysis.

7. Line 196-208: do the authors have references supporting the methodology to analyze the qualitative data?

We have updated the methodology procedure section to include references supporting the methodologies used, inductive analysis and thematic analysis, to analyze the qualitative data (232-235 tracked changes manuscript/224-228 clean copy).

8. The authors report on observational quantitative and qualitative data, did the authors consider the use of reporting guidelines such as STROBE and/or SRQR?

The reviewer correctly states that our data is both quantitative and qualitative. A formal checklist was not submitted with our research as it was not mandatory. Due to the mixed methods nature of this study, the general guidelines for SRQR and STROBE were both followed when conceptualizing the manuscript. The team can provide an official checklist if reviewers request this to be submitted.

9. The authors report quotes in the discussion, which may be better seen as results.

The authors agree with this reviewer’s recommendation and have moved most of the quotes into the results section, including Table 4.

10. The authors report several recommendations for clinical care based on their results. Given the small N and the observational nature of the data, the authors should tone down their tone and present their findings in the context of broader data. For example, while the importance of post-DBS ERP is intuitive and reported in the qualitative data, more controlled data did not unambiguously show additional effects (e.g. Graat et al., Psych Med, 2022).

We agree with this reviewer’s statement as our population was small in this research. We went through out results and discussion and worked on the language to change the tone of our clinical recommendations per the reviewer’s comments. We are unsure if the reviewer was referencing the previously mentioned Graat et al., 2021 or a different study, as we were unable to locate the listed Graat et al., Psych Med, 2022.

11. Line 366-367: in their conclusion, the authors note “It should not be assumed, based on quantitative measures alone, that QoL has not improved or has only minimally improved in participants with OCD post-DBS.” However, the authors do not present data on the general (e.g. mean) overall improvement of QoL in their sample. The authors may consider adding this data, also including e.g. percentage of (partial) responders.

The percentage change from baseline in QoL was added to the results section to address this point. The team also attached the supporting information file that includes a summary sheet with days post stimulation, Y-BOCS and QoL percent change values. Lines 255-258 (tracked changes manuscript) or 245-248 (clean copy).

Reviewer #2:

Abstract:

Authors should clarify that analyses were completed within each participant, using solely their data points, in order to assess for individual correlations

• This was addressed in the “methods” section of the abstract.

Introduction:

The introduction, while it covers relevant information, is a bit choppy, with some very short (i.e. 2 sentence) paragraphs

• The introduction was reviewed and paragraphs were lengthened to not be as short.

The final hypothesis should be clarified- “Patients will report similar improvements in QoL themes.” Were there particular themes that the author anticipated would improve versus others?

• More details were added to this hypothesis that indicate the authors anticipated themes of improvement in OCD and QoL (Lines 129-130 tracked changes manuscript/123-125 clean copy).

Methods:

It should not be necessary to include specific numerical dates for initial assessments, date of ethics determination, closure date, and date of acquisition of quantitative data, particularly given the small sample size. Either year or month and year (without date) should be sufficient.

• Per this reviewer’s suggestions, the dates for the above were simplified to month and year.

Authors should not include the coding system for their deidentification, as this is providing information to a global audience that theoretically could lead to patient identification.

• The specific coding system for deidentification was removed.

On page 6, “quantitative assessments and reviews of this data” should be “quantitative assessments and review of these data.”

• This has been corrected.

The method and related IRB approvals is unclear. If this is a retrospective analysis of already-collected clinical data, only a PHI waiver from the IRB should be necessary. However, it seems as if the qualitative analysis was added on and NOT part of the clinical procedures, just research. Thus, patients should be consented for this. The writing is not clear- it first states that no ethical approval was needed but then states that participants provided informed consent. Please clarify.

• The writers edited this piece of the “Methods” section to clarify that the study was deemed exempt. The quantitative data was part of clinical procedures and did not require informed consent per the IRB’s decision. However, the IRB required patients to provide informed consent for the additional qualitative interview data collection, as well as the use of the quantitative data for research and publication. Lines 166-172 (tracked changes manuscript) or 161-167 (clean copy).

Given that individual correlations were conducted, how many data points (range across patients) were in the correlations?

• In response to a comment by R1 (#6) we have updated the analysis approach to the relationship between mood and OCD symptoms. We have replaced individual Pearson correlations with a linear mixed model that also attempts to address a confound related to the temporal relationship of the data. Briefly, using this approach we confirmed a significant association between symptom reduction and quality of life improvement (β= -0.15, p < .001), demonstrating that the relationship remains robust after accounting for the repeated measures structure and autocorrelation. We have updated the Methods and Results sections to reflect this more rigorous analysis. Lines 219-223 tracked changes manuscript/212-216 clean copy.

Results:

Given that there is an extended period of follow-up per patient after DBS, authors should describe time line of improvements in YBOCS score and QoL For example, on page 9, authors note “Eight of ten participants reported that life improved following the DBS procedure.” What is the time frame on this? Did it persist? Were improvements in QoL and DBS based on last follow-up, any time point, etc?

• At the time of the interview, eight of ten participants reported that life improved following the DBS procedure. The timeframe between initial DBS procedure and this interview varied between individual participants However, we added additional information from the qualitative data that speaks to when participants noticed the changes and whether or the changes were long lasting for them. Table 3 was also added to provide a summary of participant responses about timeframe and longevity of impact. The team also attached the supporting information file that includes a summary sheet with days post stimulation, Y-BOCS and QoL percent change values.

Authors randomly put in that there were no changes in vision or dizziness following DBS- was that part of the QoL measure, or a statement about side effects in general?

• Vision and dizziness are measured on the Q-LES-Q-SF, which was the measure used to assess QoL. However, for clarity and conciseness, authors removed this from the results section as it did not add to the writing.

Were the timelines of changes in YBOCS and QoL different after surgery? We may anticipate that recognition of QoL may be somewhat delayed as compared to symptom improvement, which may impact correlations. Was this examined?

• At the time of the interview, eight of ten participants reported that life improved following the DBS procedure. The timeframe between initial DBS procedure and this interview varied between individual participants. The team attached the supporting information file that includes a summary sheet with days post stimulation, Y-BOCS and QoL percent change values. Figure 4 was added to capture the rate of change. The rate of change in either metric is quite variable and is captured in the quartile plot (and individual subject change over time plots). However, Figure 4 shows a higher resolution plot (30 day medians)(B). We also included a representative subject (subject 7) that the QoL is more variable than YBOCS (A). Table 3 was also added to provide qualitative data regarding timelines of change as well as long-lasting impact of DBS.

Discussion:

Authors note “Patients may retrospectively recognize that their impaired mental state…post-DBS.” Are they indicating that there are ethical concerns about these patients receiving DBS? This is what this sounds like in reading this sentence.

• The cited studies [41, 52-56] after this sentence discuss a patient’s impaired mental state pre-surgery as a potential ethical concern in regard to their ability to fully understand the outcome of DBS. Previous research has not shown substantial data that this would be an ethical concern regarding consent. The words “impaired”, “risks” and “benefits” were removed to change the tone of the sentence. Lines 409-411 (tracked changes manuscript) or 386-387 (clean copy).

1. Acevedo N, Castle DJ, Bosanac P, Groves C, Rossell SL. Patient feedback and psychosocial outcomes of deep brain stimulation in people with obsessive–compulsive disorder. J Clin Neurosci. 2023;112:80–5. doi:10.1016/j.jocn.2023.04.012.

2. Maier, F., Lewis, C. J., Horstkoetter, N., Fink, G. R., Timmermann, L., & Faust, P. (2013). Patients’ expectations of deep brain stimulation, and subjective perceived outcome related to clinical measures in Parkinson's disease: A mixed-method approach. Journal of Neurology, Neurosurgery & Psychiatry, 84(11), 1273–1281. https://doi.org/10.1136/jnnp-2012-303670

3. Hasegawa, H., Samuel, M., Douiri, A., & Ashkan, K. (2014). Patients' expectations in subthalamic nucleus deep brain stimulation surgery for Parkinson disease. World Neurosurgery, 82(6), 1295–1299.e2. https://doi.org/10.1016/j.wneu.2014.02.001

4. Thomson, C., & Carter, A. (2020). Ethical issues in experimental treatments for psychiatric disorders: Lessons from deep brain stimulation. Translational Issues in Psychological Science, 6(3), 240–246. https://doi.org/10.1037/tps0000267

5. Acevedo, N., Castle, D., Groves, C., Bosanac, P., Mosley, P. E., & Rossell, S. (2022). Clinical recommendations for the

---

## [Decision Letter · Decision Letter 1]

31 Mar 2026

PONE-D-25-56441R1Beyond the Numbers: Quantitative and Qualitative Analysis of Quality of Life After Deep Brain Stimulation for OCDPLOS One

Dear Dr. Hemendinger,

Thank you for submitting your manuscript to PLOS ONE. After careful consideration, we feel that it has merit but does not fully meet PLOS ONE’s publication criteria as it currently stands. Therefore, we invite you to submit a revised version of the manuscript that addresses the points raised during the review process. Please see the remaining issues to be addressed listed below.

We look forward to receiving your revised manuscript.

Kind regards,

Clare Eddy

Academic Editor

PLOS One

Journal Requirements:

Reviewers' comments:

Reviewer's Responses to Questions

**Comments to the Author**

1. If the authors have adequately addressed your comments raised in a previous round of review and you feel that this manuscript is now acceptable for publication, you may indicate that here to bypass the “Comments to the Author” section, enter your conflict of interest statement in the “Confidential to Editor” section, and submit your "Accept" recommendation.

Reviewer #1: (No Response)

2. Is the manuscript technically sound, and do the data support the conclusions?

Reviewer #1: Partly

3. Has the statistical analysis been performed appropriately and rigorously? 

Reviewer #1: Yes

4. Have the authors made all data underlying the findings in their manuscript fully available?

Reviewer #1: Yes

5. Is the manuscript presented in an intelligible fashion and written in standard English?

Reviewer #1: Yes

6. Review Comments to the Author

Reviewer #1: The authors have made substantial revisions to the manuscript which clearly improved it. Some remaining issues could benefit from further improvement. I list them point-by-point, referring to the original comment numbering.

3. Could the authors include the explanation on the calculation of the baseline Y-BOCS in the manuscript?

5. I still think figure 2 is hard to interpret, because strongly different durations of follow-up are compared. In addition, I don’t understand how the number of days in the quartiles can differ within a subject. I strongly suggest to make more thorough changes in this figure, e.g. in line with the previous suggestion.

10. The reference can be found here: Cognitive behavioral therapy in patients with deep brain stimulation for obsessive-compulsive disorder: a matched controlled study | Psychological Medicine | Cambridge Core

7. PLOS authors have the option to publish the peer review history of their article (what does this mean?). If published, this will include your full peer review and any attached files.

Reviewer #1: **Yes:** Roel Mocking

---

## [Author Response · Author response to Decision Letter 2]

7 Apr 2026

Thank you for your feedback and careful review of our manuscript. We have completed all revisions and our responses to each can be found below. These are the updated revisions that were provided to our team after our February 2026 submission.

Editor

• A letter that responds to each point raised by the academic editor and reviewer(s). You should upload this letter as a separate file labeled 'Response to Reviewers'. Completed

• A marked-up copy of your manuscript that highlights changes made to the original version. You should upload this as a separate file labeled 'Revised Manuscript with Track Changes'. Completed

• An unmarked version of your revised paper without tracked changes. You should upload this as a separate file labeled 'Manuscript'. Completed

Journal Requirements:

1. If the reviewer comments include a recommendation to cite specific previously published works, please review and evaluate these publications to determine whether they are relevant and should be cited. There is no requirement to cite these works unless the editor has indicated otherwise. Completed.

2. Please review your reference list to ensure that it is complete and correct. If you have cited papers that have been retracted, please include the rationale for doing so in the manuscript text, or remove these references and replace them with relevant current references. Any changes to the reference list should be mentioned in the rebuttal letter that accompanies your revised manuscript. If you need to cite a retracted article, indicate the article’s retracted status in the References list and also include a citation and full reference for the retraction notice. Completed. All references have been checked and as of 4/8/2026, there are no retracted or incorrect references that the team found upon multiple reviews and checks. References that were in the incorrect citation format were corrected.

Reviewer #1:

3. Could the authors include the explanation on the calculation of the baseline Y-BOCS in the manuscript?

Baseline Y-BOCS was defined as the mean of all available pre-DBS assessments to provide a representative estimate of pre-surgical symptom severity. In treatment-refractory OCD, symptom severity can fluctuate over time due to intrinsic variability and ongoing treatment adjustments, and a single pre-operative timepoint may reflect a transient clinical state rather than typical illness severity. Averaging across multiple pre-operative measurements reduces the influence of short-term variability. This clarification has been added to the manuscript in the methods section.

5. I still think figure 2 is hard to interpret, because strongly different durations of follow-up are compared. In addition, I don’t understand how the number of days in the quartiles can differ within a subject. I strongly suggest to make more thorough changes in this figure, e.g. in line with the previous suggestion.

Figure 2 was revised further. We thank the reviewer for this additional feedback regarding figure clarity. We have revised Figure 2 in the following aspects. 1) We removed the quartiles and represented each subject with their metric at each assessment as a function of months post stimulation onset. 2) We removed the change in increase or decrease in the metric based on color change. 3) We have added linear least square regression lines for each metric across subjects (up to 72 months).

10. The reference can be found here: Cognitive behavioral therapy in patients with deep brain stimulation for obsessive-compulsive disorder: a matched controlled study | Psychological Medicine | Cambridge Core

We thank the reviewer for providing the reference information. This reference has been added to the discussion section (line 366 in clean manuscript, line 385 in tracked changes manuscript)

---

## [Decision Letter · Decision Letter 2]

5 May 2026

PONE-D-25-56441R2Beyond the Numbers: Quantitative and Qualitative Analysis of Quality of Life After Deep Brain Stimulation for OCDPLOS One

Dear Dr. Hemendinger,

Thank you for submitting your manuscript to PLOS ONE. After careful consideration, we feel that it has merit but does not fully meet PLOS ONE’s publication criteria as it currently stands. Therefore, we invite you to submit a revised version of the manuscript that addresses the points raised during the review process.

The reviewer has highlighted a couple of minor points that remain to be addressed, shown below.

We look forward to receiving your revised manuscript.

Kind regards,

Clare Eddy

Academic Editor

PLOS One

Journal Requirements:

Reviewers' comments:

Reviewer's Responses to Questions

**Comments to the Author**

1. If the authors have adequately addressed your comments raised in a previous round of review and you feel that this manuscript is now acceptable for publication, you may indicate that here to bypass the “Comments to the Author” section, enter your conflict of interest statement in the “Confidential to Editor” section, and submit your "Accept" recommendation.

Reviewer #1: (No Response)

2. Is the manuscript technically sound, and do the data support the conclusions?

Reviewer #1: Partly

3. Has the statistical analysis been performed appropriately and rigorously? 

Reviewer #1: Yes

4. Have the authors made all data underlying the findings in their manuscript fully available?

Reviewer #1: Yes

5. Is the manuscript presented in an intelligible fashion and written in standard English?

Reviewer #1: Yes

6. Review Comments to the Author

Reviewer #1: The authors have further improved the manuscript. One final point remains which seems not fully covered by the revisions made: the figures.

In the revised manuscript, both figure 1 en 2 show the trajectories of individual Y-BOCS and QLES scores over time. But the trajectories of the lines seem quite different between the figures, how could that be explained, given that they are from the same individuals?

Moreover, the revised figure 4 is less confusing, but it remains unclear in the figure that the number of participants during follow-up decreases (which influences the IQR). This should be noted in the figure legend.

7. PLOS authors have the option to publish the peer review history of their article (what does this mean?). If published, this will include your full peer review and any attached files.

Reviewer #1: **Yes:** Roel Mocking

---

## [Author Response · Author response to Decision Letter 3]

7 May 2026

Reviewer #1: The authors have further improved the manuscript. One final point remains which seems not fully covered by the revisions made: the figures.

1. In the revised manuscript, both figure 1 and 2 show the trajectories of individual Y-BOCS and QLES scores over time. But the trajectories of the lines seem quite different between the figures, how could that be explained, given that they are from the same individuals?

The Figures represent the same information in slightly different formats. We appreciate that the inclusion of both figures may be confusing, so we have decided to Remove Figure 1.

2. Moreover, the revised figure 4 is less confusing, but it remains unclear in the figure that the number of participants during follow-up decreases (which influences the IQR). This should be noted in the figure legend.

We will update the Figure legend to note that dot size conveys number of subjects which will influence IQR.

---

## [Editor Report · Decision Letter 3]

12 May 2026

Beyond the Numbers: Quantitative and Qualitative Analysis of Quality of Life After Deep Brain Stimulation for OCD

PONE-D-25-56441R3

Dear Dr. Hemendinger,

We’re pleased to inform you that your manuscript has been judged scientifically suitable for publication and will be formally accepted for publication once it meets all outstanding technical requirements.

Kind regards,

Clare Eddy

Academic Editor

PLOS One
---

## [Editor Report · Acceptance letter]

PONE-D-25-56441R3

PLOS One

Dear Dr. Hemendinger,

I'm pleased to inform you that your manuscript has been deemed suitable for publication in PLOS One. Congratulations! Your manuscript is now being handed over to our production team.

Kind regards,

on behalf of

Dr. Clare Eddy

Academic Editor

PLOS One